# *Cyperus rotundus* Extract and Its Active Metabolite α-Cyperone Alleviates Paclitaxel-Induced Neuropathic Pain via the Modulation of the Norepinephrine Pathway

**DOI:** 10.3390/metabo14120719

**Published:** 2024-12-20

**Authors:** Keun-Tae Park, Insuk Sim, Jae-Chul Lee, Young-Ho Jin, Woojin Kim

**Affiliations:** 1Department of Physiology, College of Korean Medicine, Kyung Hee University, Seoul 02453, Republic of Korea; cerex@naver.com (K.-T.P.); crewera@gmail.com (J.-C.L.); 2Korean Medicine-Based Drug Repositioning Cancer Research Center, College of Korean Medicine, Kyung Hee University, Seoul 02453, Republic of Korea; 3Department of Clinical Laboratory Science, Dongnam Health University, Suwon 16328, Republic of Korea; simis0519@dongnam.ac.kr; 4Department of Physiology, School of Medicine, Kyung Hee University, Seoul 02453, Republic of Korea

**Keywords:** chemotherapy-induced neuropathic pain, *Cyperus rotundus*, α-cyperone, paclitaxel

## Abstract

Background: Paclitaxel is a widely used anticancer drug for ovarian, lung, breast, and stomach cancers; however, its clinical use is often limited by the side effects of peripheral neuropathy. This study evaluated the effects of *Cyperus rotundus* (*C. rotundus*) extract and its active metabolite, α-cyperone, on paclitaxel-induced neuropathic pain. Methods: The oral administration of *C. rotundus* extract at doses of 500 mg/kg and intraperitoneal administration of α-cyperone at doses of 480 and 800 μg/kg prevented both the development of cold and mechanical pain. Results: The gene and protein expressions of tyrosine hydroxylase and noradrenergic receptors (α1- and α2-adrenergic), which were upregulated by paclitaxel, were significantly downregulated in the *C. rotundus* extract-treated group. In the locus coeruleus region of the mouse brain, *C. rotundus* extract administration also reduced the elevated expression of tyrosine hydroxylase induced by paclitaxel. The concentration of α-cyperone in *C. rotundus* extract was quantified using high-performance liquid chromatography (HPLC). In the group treated with α-cyperone, at levels corresponding to its content in *C. rotundus*, both cold and mechanical allodynia were effectively prevented. Conclusions: This study suggests that α-cyperone shows potential as a preventive agent for paclitaxel-induced neuropathic pain.

## 1. Introduction

Pain is a global healthcare problem that has detrimental effects on the quality of life and causes a high economic burden on individuals and countries [1,2]. Neuropathic pain is caused by damage to the central nervous system due to disease states (e.g., stroke, diabetes, and multiple sclerosis), toxic issues (e.g., chemotherapy), or lesions (e.g., trauma or diagnosis abnormalities) [3,4]. A clear goal in the treatment of neuropathic pain is the prevention of disease onset and the control of established neuropathic pain mechanisms [5]. Current treatments for neuropathic pain do not provide satisfactory antinociceptive effects due to limited dosage, efficacy, and side effects, and ongoing research on antinociceptive is continuously needed [1,6,7]. We performed a study on the antinociceptive effects of morphine, bee venom, and medicinal herbs on neuropathic pain [8,9]. Herbal medicine is advantageous in the development of pain relievers because it has fewer side effects and is relatively safe compared to synthetic drugs [10,11].

Paclitaxel is a chemotherapy drug extracted and developed from the tree *Taxus brevifolia*, and it is used worldwide for ovary, lung, and breast cancer [12,13,14]. Although it is highly effective, its use and the treatment period are limited due to the major side effect of paclitaxel, peripheral neuropathic pain [15]. Numbness of the extremities, cold allodynia, and mechanical allodynia occur within 24 h after the first administration [16], and these side effects lead to the discontinuation of chemotherapy and a decrease in the patient’s quality of life [17]. Therefore, the development of a drug that can effectively alleviate chemotherapy-induced neuropathic pain with fewer side effects is needed [18].

*Cyperus rotundus* is a plant in the *Cyperaceae* family, widely distributed in the Mediterranean Basin [19]. Its rhizome (*Cyperus rhizoma*), widely used for medicinal purposes, is described in many national pharmacopeias, including Korean Pharmacopoeia [20]. It is used as a traditional medicinal plant in India, China, and Japan, for the treatment of inflammatory diseases, gastrointestinal disorders, and spasms [21,22,23]. Other pharmacological studies have reported that it has antipyretic effects [24]. According to Ayurveda, *C. rotundus* is considered to be a diaphoretic, carminative, tonic, insect repellent, analgesic, and gastrointestinal and low blood pressure remedy [25]. Phytochemical analysis revealed that *C. rotundus* contains essential oils, saponins, alkaloids, and polyphenols [26,27].

Various active ingredients were identified in *C. rotundus*, including α-cyperone, cyperene, cineole, β-selinene, sugeonol, patchoulenone, sesquiterpenes, oleanolic acid, and glycerol [28,29]. The main ingredient of *C. rotundus*, α-cyperone, is contained in very high concentrations and suppresses LPS-induced COX-2 expression in RAW264.7 cells, inhibiting the production and mRNA expression of IL-6, and inflammatory cytokine. In addition, α-cyperone treatment in LPS-induced RAW 264.7 cells inhibited the transcriptional activity of NFκb and the nuclear translocation of the p65NFκB subunit [30].

In this study, we attempted to secure the analgesic effect and mechanism for chemotherapy-induced neuropathic pain based on the traditional efficacy of *C. rotundus* rhizome mentioned above. In addition, we tested the hypothesis that the administration of *C. rotundus* and its metabolite α-cyperone would prevent and suppress paclitaxel-induced chemotherapy-induced peripheral neuropathic pain by modulating the norepinephrine (NE) analgesic system. We modified the timing of *C. rotundus* administration to examine the efficacy of pharmacological treatment in preventing the development of paclitaxel-induced neuropathic pain and suppressing the maintenance of established neuropathic pain. Through this hypothesis, we estimated that *C. rotundus* and α-cyperone could suppress paclitaxel-induced neuropathic pain via NE modulation, and we aimed to provide a foundation for the development of natural product-based analgesic agents.

## 2. Methods

### 2.1. Animals

Eight-week-old C57BL/6 mice purchased from Daehan Biolink (DBL, Chung-Buk, Republic of Korea) and housed in plastic cages under optimal conditions (23 °C ± 2 °C; 65 ± 5% humidity; 12 h light and 12 h dark cycle). All animals were acclimated to the cages for 1 week prior to the study. The animal experiments were approved by the Institutional Animal Care and Use Committee of Kyung Hee University (KHUASP(SE)-23-223, 7 August 2024). At the end of the study, animals were anesthetized and sacrificed by inhaling excessive isoflurane.

### 2.2. Administration of Paclitaxel, C. rotundus and α-Cyperone

Paclitaxel (Sigma Aldrich, St Louis, MO, USA) was dissolved in a 1:1 mixture of ethanol and Cremophor at a concentration of 6 mg/mL. Paclitaxel was diluted in phosphate-buffered saline (PBS) to a final concentration of 0.2 mg/mL prior to injection into mice. PBS without paclitaxel was injected as a control group. Paclitaxel (2 mg/kg) and vehicle were injected intraperitoneally (D0, D2, D4, D6) a total of four times, and the total cumulative dose was 8 mg/kg. *C. rotundus* samples were diluted in PBS at doses of 100, 300, and 500 mg/kg and administered orally using oral sonde (Jungdo BNP, Seoul, Republic of Korea). α-Cyperone was administered according to the α-cyperone analysis content in the *C. rotundus* administration group. α-Cyperone was initially dissolved in DMSO at 10 mg/mL, and for intraperitoneal administration, it was prepared by dissolving it in PBS at concentrations of 160, 480, and 800 μg/kg. *C. rotundus* and α-cyperone were administered on D0, D2, D4, and D6 (Figure 1A).

### 2.3. C. rotundus Extract Preparation

The herbal medicine used in this study was *Cyperus rhizome*, and the identity of plant species (*C. rotundus*) was confirmed through ITS gene analysis (Appendix A). Dried *C. rotundus* was provided by Young-cheon medicinal herb Co., Ltd. (Young-cheon, Republic of Korea). *C. rotundus* extract was prepared using the reflux method with 30% ethanol for 6 h at 80 °C. The extract was filtered and concentrated using an evaporator under reduced pressure. *C. rotundus* was dried with an evaporator and lyophilized using a free dryer. The *C. rotundus* extract was diluted in PBS and administered orally to mice at an equal volume of 0.1 mL for each concentration.

### 2.4. Behavioral Assessments

Cold and mechanical allodynia were assessed using the acetone drop and von Frey filaments test methods. In the acetone drop method, 10 μL of acetone was spread onto the paw of mice, and the degree of pain was assessed. In brief, mice were acclimated in the behavioral assessment chamber for 30 min before the behavioral assessment. Acetone was sprayed onto both hind paws using a pipette. After spraying, the number of licking, flinching, and withdrawals was counted for 30 s. The ‘# Response’ in the results represents the average number of responses for each group.

Mechanical allodynia was assessed using von Frey filaments (Stoelting, Chicago, IL, USA). Von Frey filaments of different stiffness levels (2, 1.4, 1, 0.6, 0.4, 0.16, 0.07, 0.04, and 0.02 g) were inserted into the mid-plantar hind paw. The ‘50% threshold’ of the results was determined by Chaplan and Dixon. Calculation was performed according to the OX method (up and down). After behavioral evaluation, mice were anesthetized with isoflurane and euthanized by cardiac perfusion with PBS. Lumbar 4–5 spinal cord segments were sampled and further analyzed.

### 2.5. mRNA Expression by Quantitative Real-Time Polymerase Chain Reaction (qRT-PCR)

Ribonucleic acid (RNA) was extracted from spinal tissues and isolated by using an RNA extraction kit (AccuPrep, Bioneer, Daejeon, Republic of Korea), according to the user’s manual. The concentration of RNA was measured using a NanoDrop spectrophotometer (Thermo Scientific, Middlesex, MA, USA). Complementary DNA (cDNA) was synthesized from the extracted RNA using a Maxime RT kit (Intronbi, Seongnam, Republic of Korea). The real-time PCR was performed by using a Sensi FAST SYBR kit (Meridian Bioscience, Cincinnati, OH, USA) and a CFX Real-Time PCR Detection System (Bio-Rad Laboratories, Hercules, CA, USA). The primer used for the PCR were as follows: α2-adrenergic receptor (*Adra2a*) forward 5′-AAA CCT CTT CCT GGT GTC TC-3′; α1-adrenergic receptor (*Adra1a*) forward 5′-ATG CTC CAG CCA AGA GTT CA-3′ and reverse 5′-TCC AAG AAG AGC TGG CCT TC-3′; tyrosine hydroxylase (TH) forward 5′GACATTGCCCAGAGATGCAAGTCCAATGTC-3′ and reverse 5′-TGTTGGCTGACCGCACATTT-3′; *Gapdh* forward 5′-GGA GGT AGC TCC TGA TTC GC-3′ and reverse 5′-CAC ATT GGG GGT AGG AAC AC-3′. The reaction was preheated for 10 min at 95 °C followed by 40 cycles at 95 °C for 20 s, 58 °C for 20 s, and 72 °C for 20 s. GAPDH was used as a standard to quantify the amount of RNA (0.2 μg) in each sample. The qPCR data for amplification-produced fluorescence were used to calculate a specific detection threshold (Ct value). The relative quantification of each group was calculated using the delta–delta Ct (_ΔΔ_Ct) calculation method, and the expression value of the control group was expressed as 1.

### 2.6. Protein Expression by Western Blot

Spinal cord tissues were homogenized using the radio-immunoprecipitation assay (RIPA) buffer and centrifuged at 13,500 rpm for 10 min. Protein quantification was determined using the protein assay kit (Bradford, Bio-Rad Laboratories, Hercules, CA, USA). Protein samples were then loaded to 15% sodium dodecyl sulfate–polyacrylamide gel electrophoresis and then transferred to a nitrocellulose membrane at 110V for 100 min. The transferred nitrocellulose membrane was blocked with 5% skim milk for 1.5 h. Afterward, the nitrocellulose membranes were washed with Tris-buffered saline with tween 20 (TBS-T) and incubated with primary antibodies overnight at 4 °C. α1-Adrenergic receptor, α2-adrenergic receptor, tyrosine hydroxylase, and actin were detected with rabbit polyclonal anti-α1-adrenergic receptor (dilution 1:3000, #NB100-78585, Novus Biologicals, Littleton, MA, USA), rabbit polyclonal anti-α2-adrenergic receptor (dilution 1:1000, #NB2-66606, Novus Biologicals, Littleton, CO, USA), rabbit polyclonal anti-tyrosine hydroxylase (dilution 1:1000, #NB2-66606, Novus Biologicals, Littleton, CO, USA), and rabbit polyclonal anti-actin (dilution 1:1000, #NB100-1617, Novus Biologicals, Littleton, CO, USA), respectively. They were further incubated with secondary antibody (HRP conjugated anti-rabbit antibody (#31460) at a 1:10,000, Thermo Scientific, Waltham, MA, USA) for 1 h at room temperature. Protein bands were detected with an enhanced chemiluminescence solution (D-Plus ECL Femto System, Hwang-Seong, Republic of Korea).

### 2.7. Immunohistochemistry (IHC)

Mice were anesthetized with isoflurane, and the brain tissues were fixed with 4% paraformaldehyde (PFA). After transcardial perfusion, the brain tissues were collected and post-fixed with 4% PFA (5 h), saturated with 30% sucrose at 4 °C overnight. The tissues were embedded into OCT and frozen. The frozen tissues were cryosectioned. The sliced sections were blocked with 5% normal goat serum and 0.3% Triton-PBS for 1 h. Then, tissues were incubated with primary antibody overnight at 4 °C. After being washed with PBS and incubated with secondary antibody for 1 h at room temperature, they were mounted with DAPI NB-100-1617 (Novus Biologicals, Littleton, CO, USA) and Alexa 546 goat anti-rabbit (1:1000, A10040, Thermo Scientific, Waltham, MA, USA). The images of cross-sections were documented using a laser scanning confocal microscope (Zeiss LSM 910, Jena, Germany), capturing the areas of interest in photographs. The analysis of tyrosine hydroxylase expression in the locus coeruleus (LC) was conducted using ImageJ software (National Institute of Health, Bethesda, MD, USA, version 1.54f).

### 2.8. α-Cyperone Analysis by HPLC

HPLC analysis was performed using an Agilent Infinity ll HPLC 1260 and UV detector. α-Cyperone analysis conditions are shown in Table 1. Stock solutions from α-cyperone (120 µg/mL) were prepared in methanol. The α-cyperone used in this study was purchased from PhytoLab GmbH (Vestenbergsgreuth, Germany). Six dilutions of α-cyperone (100, 50, 25, 12.5, 6.25, and 3.125 μg/mL) were prepared and subjected to HPLC analysis. α-Cyperone solution (100 mg) was ultrasonically extracted (4 °C, 1 h) with 1 mL of 30% methanol. The diluted solution was centrifuged at 4 °C at 13,000 rpm for 10 min, and the supernatant was filtered through a 0.45 µm syringe filter to obtain the analysis solution.

## 3. Results

### 3.1. Analgesic Effect of Multiple Administration of C. rotundus on Cold and Mechanical Allodynia Induced by Paclitaxel

Three different doses of *C. rotundus* extracts (100, 300, and 500 mg/kg) were administered orally to evaluate its preventive effects on paclitaxel-induced neuropathic pain. The timeline for paclitaxel administration and behavioral assessment is shown in Figure 1A. The results indicated that *C. rotundus* at doses of 300 and 500 mg/kg significantly alleviated paclitaxel-induced cold allodynia, while a significant analgesic effect on mechanical allodynia was observed only in the 500 mg/kg group (Figure 1B,C). Overall, *C. rotundus* exhibited a more pronounced effect on cold allodynia than on mechanical allodynia, with the most potent analgesic effect observed at the 500 mg/kg dose.

### 3.2. Dose-Dependent Downregulation of Adrenergic Receptors and Tyrosine Hydroxylase by C. rotundus in Paclitaxel-Induced Neuropathic Pain

Gene expression levels of the α1-adrenergic receptor (α1ADR), α2-adrenergic receptor (α2ADR), and tyrosine hydroxylase were assessed in the spinal cord of mice treated with paclitaxel and *C. rotundus*. Tyrosine hydroxylase is a crucial enzyme for NE synthesis. In the paclitaxel-treated group, the gene expression of α1ADR, α2ADR, and tyrosine hydroxylase was significantly elevated. However, in the *C. rotundus*-treated group, these gene expressions were reduced in a dose-dependent manner (Figure 2A).

To further verify the consistency between gene expression and protein levels of spinal α1ADR, α2ADR, and tyrosine hydroxylase following paclitaxel and *C. rotundus* treatment, Western blot analysis was conducted. Tissue samples were collected from the lumbar 4–5 spinal cord segments of the mice. The results demonstrated that protein expressions of α1ADR, α2ADR, and tyrosine hydroxylase were significantly upregulated in the paclitaxel-treated group compared to the control group (Figure 2B).

### 3.3. C. rotundus Reduces Tyrosine Hydroxylase Expression in the Brain’s Locus Coeruleus in Paclitaxel-Treated Mice

Since most spinal NE is synthesized by tyrosine hydroxylase (TH) in the locus coeruleus (LC) of the brain, TH expression was assessed using an anti-TH antibody. On Day 10, the final day of the experiment, TH levels in the brains of mice treated with paclitaxel and *C. rotundus* were evaluated (Figure 3A). When the intensity of TH expression, labeled with Alexa-488, was measured, the results showed that TH expression, which was elevated in the paclitaxel-treated group (Figure 3C) compared to the control group (Figure 3B), was significantly reduced in the *C. rotundus* 300 mg/kg (Figure 3D) and 500 mg/kg (Figure 3E) groups. The intensity of the TH fluorescence signal is quantified in Figure 3F.

### 3.4. Qualification and Quantification of α-Cyperone in C. rotundus Using High-Performance Liquid Chromatography (HPLC)

HPLC analysis was performed to quantify α-cyperone, an active compound in *C. rotundus* extract. The retention time for α-cyperone in the standard solution was approximately 8.8 min (Figure 4A). The UV spectrum and retention time of α-cyperone in the *C. rotundus* extract sample were precisely consistent with those of the standard (Figure 4B). The calibration curve exhibited linearity across the concentration range of α-cyperone (120, 60, 30, 15, and 7.5 µg/mL), with a regression equation of y = 4.69947x + 27.28663 and an R^2^ value of 0.99994, indicating high accuracy and reliability in the quantification method.

Based on HPLC analysis, the concentration of α-cyperone in *C*. *rotundus* was determined to be approximately 0.161%. Specifically, 1.61 mg of α-cyperone was detected per 1000 mg of *C. rotundus* extract, yielding a calculated content of (1.61 mg/1000 mg) × 100 = 0.161%.

### 3.5. Analgesic Effect of α-Cyperone in Cold and Mechanical Allodynia Induced by Paclitaxel

As demonstrated by the HPLC analysis results, α-cyperone was identified as a component of *C. rotundus*. Therefore, cold and mechanical allodynia behavioral assessments were performed to evaluate whether α-cyperone could actually relieve neuropathic pain with paclitaxel. α-Cyperone was administered intraperitoneally to mice at doses of 160, 480, and 800 μg/kg. The doses were selected according to the results of HPLC analysis. *C. rotundus* contained 0.161% α-cyperone, which was administered at concentrations of 160 μg/kg (100 mg/kg × 0.161% = 160 μg/kg, α-cyperone low dosage, ACL), 480 μg/kg (300 mg/kg × 0.161% = 480 μg/kg, α-cyperone medium dosage, ACM), and 800 μg/kg (500 mg/kg × 0.161% = 800 μg/kg, α-cyperone high dosage, ACH). Von Frey filament test and acetone drop were used to assess mechanical and cold hypersensitivity in mice, respectively. In assessing cold and mechanical allodynia, α-cyperone showed significant analgesic effects in a concentration-dependent manner in ACM and ACH (Figure 5A,B). However, the analgesic effect of α-cyperone was better at relatively low concentrations in cold allodynia than in mechanical allodynia.

### 3.6. Effect of C. rotundus on Akt Phosphorylation Expression

Phosphorylated Akt (pAkt) has been reported to be increased in various pain conditions and is closely related to the development of chronic pain, suggesting that Akt phosphorylation may contribute to pain signal transduction [31]. pAkt is upregulated upon administration of paclitaxel, an anticancer drug that induces neuropathic pain [32]. In addition, pAkt has been reported to be activated and increase protein synthesis via the PI3K, mTOR/S6K, and MAPK cascade by NE treatment [33].

In this study, the protein expression of pAkt was examined, which is a key intermediary influenced by paclitaxel and NE. In the paclitaxel-treated group, pAkt levels were significantly elevated, which may contribute to the pain state through Akt phosphorylation. However, in the *C. rotundus*-treated group, Akt phosphorylation was significantly suppressed compared to the paclitaxel group (Figure 6A,B), suggesting that *C. rotundus* may exert its analgesic effects in part by modulating the Akt signaling pathway. Thus, it can be hypothesized that inhibition of Akt phosphorylation may contribute to pain relief by attenuating pain signal transmission within the spinal cord.

## 4. Discussion

In this study, we investigated the analgesic effect of *C. rotundus* on paclitaxel-induced neuropathic pain and explored its mechanism. *C. rotundus* was administered orally at concentrations of 100, 300, and 500 mg/kg, respectively, and it was confirmed that *C. rotundus* relieved cold and mechanical allodynia induced by paclitaxel. In addition, the expression of the NE gene and protein in the spinal cord was increased by paclitaxel, but NE expression was decreased along with analgesia due to *C. rotundus* administration. The content of α-cyperone, the main substance of *C. rotundus*, was analyzed by HPLC and was 0.161%, and similar analgesic effects were observed when administered intraperitoneally at the different concentrations of the extract.

In a previous study, *C. rotundus* administered at 100 and 200 mg/kg showed significant analgesic effects in the hot plate test and the tail immersion test in formalin-induced pain [34], and when administered at 200 mg/kg in mice with carrageenan-induced edema and inflammatory pain, it showed the highest analgesic effects in the anti-edema and licking tests [35]. Several studies have reported that most *C. rotundus* toxicity studies are safe [36], and the lethal dose LD50 when administered intraperitoneally to mice was 90 g/kg [37]. However, this study is the first to report the analgesic effect of *C. rotundus* on neuropathic pain caused by the anticancer drug paclitaxel. It was confirmed that the repeated administration of *C. rotundus* could treat and prevent cold and mechanical allodynia induced by paclitaxel injection four times in a dose-dependent manner, and overall, *C. rotundus* showed a higher analgesic effect on cold allodynia than on mechanical allodynia.

The involvement of the spinal NE system as the underlying mechanism of analgesic action has been demonstrated. In this study, to determine the relevance of the serotonergic system, the gene expression of 5-HT_1A_ and 5-HT_3A_ receptors in the spinal cord was checked, and there was no significant difference (Appendix A). Multiple paclitaxel administration doses induced an increase in the spinal TH, as well as α1- and α2-adrenergic receptors. In addition, the positive cell intensity of TH was upregulated in the LC, a neuron belonging to the A6 noradrenergic group. The functions of α1- and α2-adrenergic receptors have been reported extensively in neuropathic pain studies [38,39], and the intraperitoneal injection of these adrenergic receptor antagonists significantly alleviated hypersensitivity in rats with nerve injury-induced neuropathic pain, suggesting that both receptors are involved in pain enhancement and are candidates for novel analgesic agents [40].

In general, NE release is known to have an analgesic effect in pain situations. Many drugs, such as clonidine, NE reuptake inhibitors, and gabapentinoids, are designed to treat neuropathic pain and to produce analgesia by activating, mimicking, or augmenting the descending NE pathway [41,42,43]. This suggests that the descending NE pathway is very important for endogenous analgesia and is the primary target of many drugs approved to treat neuropathic pain.

On the other hand, the long-term release of adrenaline or chronic stress can be a factor that worsens pain. When neuropathic pain turns into chronic pain, NE in the LC becomes less responsive to noxious stimuli, thereby reducing endogenous analgesia. Astrocyte glutamate dysregulation is involved in this damage and is considered a major factor [44]. The continuous activation of the sympathetic nervous system can cause muscle tension, which can lead to chronic pain or myalgia. If adrenaline is secreted continuously, it can increase the inflammatory response or cause excessive tension, which can actually make the pain more noticeable. The theoretical possibility has been raised that the use of antiadrenergic agents in patients with fibromyalgia can reduce chronic pain intensity by reducing excessive sympathetic activity [45]. Therefore, the effect of increasing NE in reducing or worsening pain may vary depending on the situation.

The descending pain modulatory system contains noradrenergic projections that support a tonic pathway from the LC to the diffuse noxious inhibitory controls [46] and the dorsal horn of the spinal cord [47]. Additionally, the LC is the main production site and source of NE in the brain and supplies NE to the brain and spinal cord [48]. TH is a rate-limiting enzyme that determines the presence and rate of the reaction in the biosynthesis of NE in the LC, and this regulatory mechanism plays an important role in controlling catecholaminergic action [49,50]. It is still not completely known whether NE secreted from the LC promotes or inhibits pain. Some studies have found that activating the LC facilitates and maintains pain. In mice with pain induced by nerve injury, the responsiveness to noxious mechanical stimuli was found to be enhanced in LC neurons [51]. Additionally, when lidocaine was administered to the LC of hyperalgesia mice, neuropathy symptoms were reduced, confirming that the function of the LC is closely related to pain [52].

In this study, α-cyperone administration at low doses (480 and 800 μg/kg) showed analgesic effects in paclitaxel-induced neuropathic pain. In addition, α-cyperone showed a dose-dependent effect, suggesting that α-cyperone is the main substance exhibiting analgesic effects. Taxanes, vinca alkaloids, and colchicine analogs interact with microtubulin, and they are distinguished by their respective binding domains [53]. Paclitaxel, a taxane, contributes to the stability of microtubules, unlike vinca alkaloids and colchicine, and thus causes neuropathic pain [54]. According to a docking analysis study of α-cyperone, it binds to the region near Taxol and interacts with amino acids Trp21, Ile24, Ser25, Gln43, Leu44, Glu47, Arg48, Ala57, Gly58, Asn59, Lys60, Tyr61, Val62, Pro63, Phe83, and Gly84 in the GTP binding site of β-tubulin [55]. In addition, it was confirmed that cyperone inhibits the growth of human cervical cancer [56]. This study revealed that α-cyperone affects the microtubule assembly and the stabilization of paclitaxel, and when α-cyperone binds to β-tublin, a structural change in the binding domain occurs, which decreases the stability of microtubulin. Accordingly, our additional hypothesis is that α-cyperone contributes to the destabilization of microtubulin and the occurrence of analgesic effects, but this requires further study.

Our results show that α-cyperone has the potential to prevent and treat paclitaxel-induced neuropathic pain because α-cyperone significantly relieved the pain and elucidated the mechanism related to the NE system. Additional necessary studies are needed to establish standardization, such as maintaining the α-cyperone content during the demand stage and the extraction process of *C. rotundus*. Furthermore, in-depth studies are needed on how α-cyperone regulates NE levels and the function of α1- and α2-adrenergic receptors.

## 5. Conclusions

Our study showed that the intraperitoneal injection of *C. rotundus* and α-cyperone significantly reduced paclitaxel-induced pain and attenuated pain-induced increased NE in the spinal cord. These results suggest that *C. rotundus* can alleviate neuropathic pain by inhibiting NE/pAkt signaling in the spinal cord. Our study provided baseline data for preclinical studies of *C. rotundus* for the treatment and prevention of paclitaxel-induced neuropathic pain.

## Figures and Tables

**Figure 1 metabolites-14-00719-f001:**
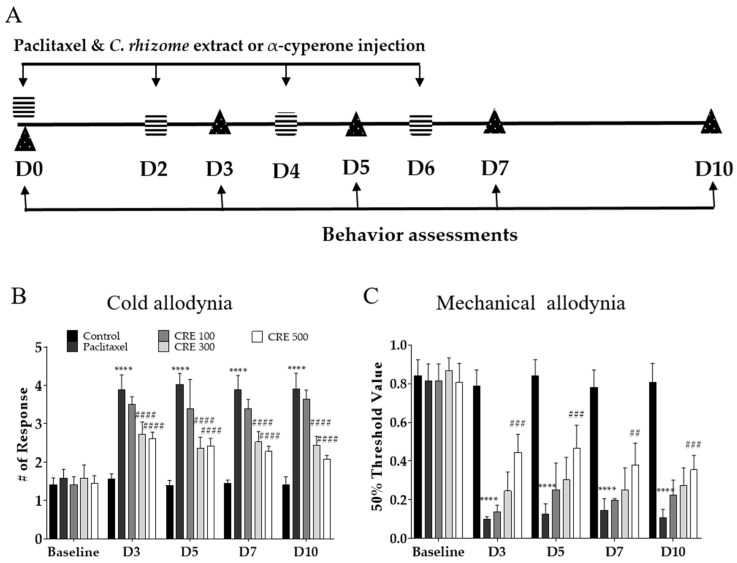
Effects of *C. rotundus* on cold and mechanical allodynia induced by paclitaxel. The experiment schedule (**A**) and the effect of *C. rotundus* on paclitaxel-induced cold (**B**) and mechanical allodynia (**C**). CRE 100, 300, and 500 are 100, 300, and 500 mg/kg of *C. rotundus*, respectively. *n* = 6 in each group. CRE represents *C. rotundus* extract. Black striped boxes represent paclitaxel and drug administration, and black triangles represent behavior assessments. The results are expressed as the mean ± standard deviation. **** *p* < 0.0001 vs. control, ## *p* < 0.01, ### *p* < 0.001, #### *p* < 0.0001 vs. paclitaxel with two-way ANOVA followed by Tukey’s post-test for multiple comparisons.

**Figure 2 metabolites-14-00719-f002:**
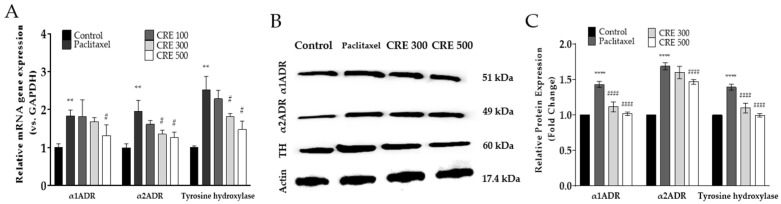
Gene and protein expression analysis of adrenergic receptor genes and tyrosine hydroxylase in the spinal cord. Effects of paclitaxel and *C. rotundus* on the gene expression of adrenergic receptors and tyrosine hydroxylase (**A**). Protein analysis was assessed using Western blot from the lumbar 4–5 level of the spinal cord. Representative image of Western blot (**B**) and analysis of relative expression levels of target proteins in the spinal cord (**C**). CRE 100, 300, and 500 are 100, 300, and 500 mg/kg of *C. rotundus*, respectively. *n* = 6 in each group. The results are expressed as the mean ± standard deviation. NS, not significant. ** *p* < 0.01, **** *p* < 0.0001 vs. control, # *p* < 0.05, #### *p* < 0.0001 vs. paclitaxel with two-way ANOVA followed by Tukey’s post-test for multiple comparisons.

**Figure 3 metabolites-14-00719-f003:**
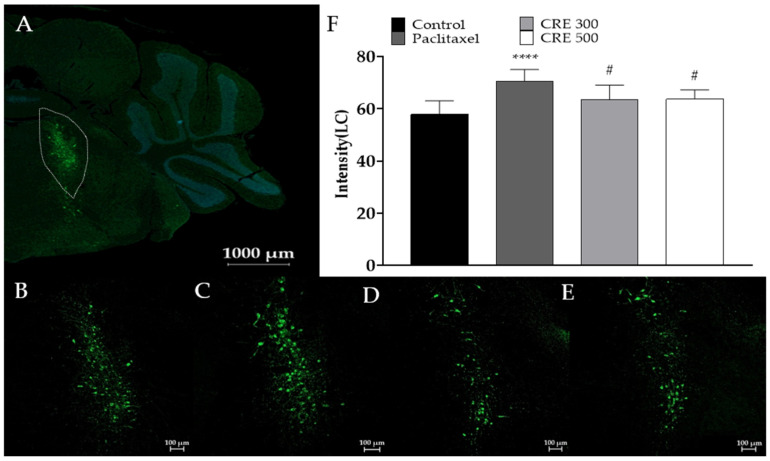
Analysis and quantification of tyrosine hydroxylase (TH) expression using immunohistochemistry in mouse brains. Brain sections were fixed and stained with the anti-TH antibody in the locus coeruleus region, and the fluorescent dye Alexa-488 fluoresces to indicate TH (**A**), control (**B**), paclitaxel (**C**), CRE 300 mg/kg (**D**), and 500 mg/kg (**E**) administration changed TH protein expression in mouse brain and intensity quantification (**F**). CRE represents *C. rotundus* extract. The area within the white dotted line represents the locus coeruleus. *n* = 6 in each group. The results are expressed as the mean ± standard deviation. NS, not significant. **** *p* < 0.0001 vs. control, # *p* < 0.05 vs. paclitaxel with two-way ANOVA followed by Tukey’s post-test for multiple comparisons.

**Figure 4 metabolites-14-00719-f004:**
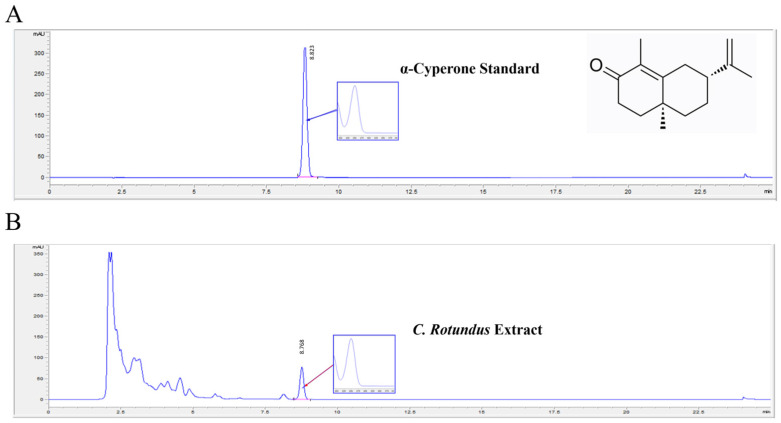
Identification and quantification of α-cyperone in the *C. rotundus* extract using high-performance liquid chromatography (HPLC). HPLC chromatograms of α-cyperone standard (**A**) and α-cyperone (**B**) in the extract. The red arrows represent the peaks of the standard, and the blue line represents α-cyperone. The X-axis reports the retention time, and the Y-axis is the absorbance unit.

**Figure 5 metabolites-14-00719-f005:**
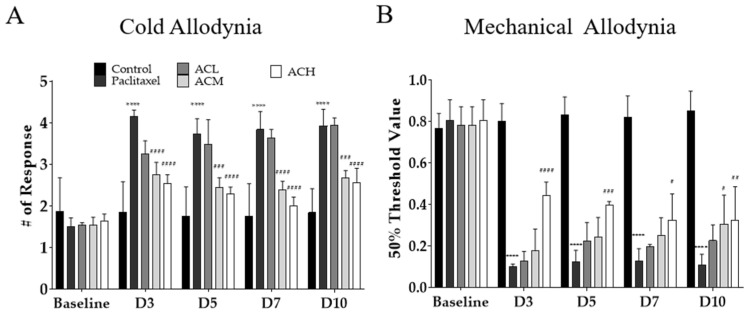
Effects of α-cyperone on cold and mechanical allodynia induced by paclitaxel. The effect of α-cyperone against paclitaxel-induced cold (**A**) and mechanical allodynia (**B**). ACL, α-cyperone low dose; ACM, α-cyperone medium dose; ACH, α-cyperone high dose. *n* = 6 in each group. The results are expressed as the mean ± standard deviation. **** *p* < 0.0001 vs. control, # *p* < 0.05, ## *p* < 0.01, ### *p* < 0.001, #### *p* < 0.0001 vs. paclitaxel with two-way ANOVA followed by Tukey’s post-test for multiple comparisons.

**Figure 6 metabolites-14-00719-f006:**
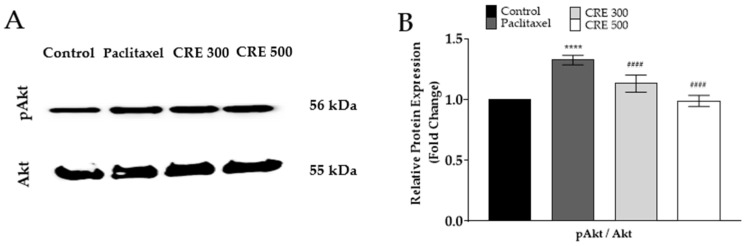
Protein quantification of Akt and pAkt in paclitaxel-induced peripheral neuropathic pain mice. Protein analysis was assessed using Western blot from the lumbar 4–5 level of the spinal cord. Representative image of Western blot (**A**) and analysis of relative expression levels of Akt, and pAkt proteins in the spinal cord (**B**). CRE 300 and 500 are 300 and 500 mg/kg of *C. rotundus*, respectively. *n* = 6 in each group. The results are expressed as the mean ± standard deviation. NS, not significant. **** *p* < 0.0001 vs. control, #### *p* < 0.0001 vs. paclitaxel with two-way ANOVA followed by Tukey’s post-test for multiple comparisons.

**Table 1 metabolites-14-00719-t001:** Analytical conditions of HPLC for α-cyperone analysis.

Conditions			
Treatment	α-Cyperone		
Column	YMC-Triart C18		
Flow rate	1.0 mL/min		
Injection volume	10 μL		
UV detection	254 nm		
Run time	25 min		
Time (min)	% Acetonitrile	Distilled Water	Flow Rate (mL/min)
0	80	20	1.0
15	80	20	1.0
17	100	0	1.0
20	100	0	1.0
21	80	20	1.0
25	80	20	1.0

## Data Availability

All the data supporting the conclusions of this study are included in the manuscript.

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
