# Peer review of "Cyperus rotundus Extract and Its Active Metabolite α-Cyperone Alleviates Paclitaxel-Induced Neuropathic Pain via the Modulation of the Norepinephrine Pathway"

_metabolites, 2024, doi:10.3390/metabo14120719_

Round 1

Reviewer 1 Report (Previous Reviewer 2)

Comments and Suggestions for Authors

Authors should:

1. The corrections made should have been highlighted through the whole manuscript. Unfortunately, only few were made, and authors did not made the other required corrections including typos and answer to some other comments.

2. Authors should write the scientific name of the plant correctly whenever mentioned in the manuscript. It should be written as: Cyperus rotundus, abbreviated as C. rotundus.  It is not acceptable to use upper case with species name even (Cyperus rotundus) even if you are using CRE as abbreviation in the whole text.

3. Authors should unify units of measurements like mL in the whole manuscript and according to the accepted styling of the format. Either use mL or ml.

4. In section 3.4, the equation is : y = 4.69947 + 27.28663; this is not an equation? its simply two numbers adding together???

moreover, authors have determined the concentration of a-cyperone as 0.161%, based on your method, you should be able to find the concentration not the relative concentration. As in the previous revision, this figure should be clarified.

5. Authors should revise carefully the list of references, previous comments were not considered and the required corrections were not made. authors should follow the Journal's accepted styling format when citing the journal title: full title or abbreviated one, italics or not, ...etc.

6. also unify the way you write subtitles, first letter of every word either upper case or lower case in all subtitles. 

Author Response

  1. The corrections made should have been highlighted through the whole manuscript. Unfortunately, only few were made, and authors did not made the other required corrections including typos and answer to some other comments.
  2. Authors should write the scientific name of the plant correctly whenever mentioned in the manuscript. It should be written as: Cyperus rotundus, abbreviated as C. rotundus. It is not acceptable to use upper case with species name even (Cyperus rotundus) even if you are using CRE as abbreviation in the whole text.

→ The exact name of C. rotundus was confirmed with the Korean Food and Drugs Administration (KFDA) and was found to be C. rhizome. All C. Rutundus were corrected to C. rhizome. They were highlighted in red in the figures and text of the manuscript.

  1. Authors should unify units of measurements like mL in the whole manuscript and according to the accepted styling of the format. Either use mL or ml.

→ It was revised to unify it into 'mL'.

  1. In section 3.4, the equation is : y = 4.69947 + 27.28663; this is not an equation? its simply two numbers adding together???

→ The formula in the text, y = 4.69947 + 27.28663, was modified to the equation, y = 4.69947x + 27.28663.

moreover, authors have determined the concentration of a-cyperone as 0.161%, based on your method, you should be able to find the concentration not the relative concentration. As in the previous revision, this figure should be clarified.

→ An explanation of the a-cyperone content of 0.161% was added to the manuscript.

→ [To elaborate on the calculation, the content of α-cyperone was analyzed as 1.61 mg in 1000 mg of initial CRE during HPLC analysis, and according to [1.61 mg / 1000 mg] x 100, 0.161% was derived.]

  1. Authors should revise carefully the list of references, previous comments were not considered and the required corrections were not made. authors should follow the Journal's accepted styling format when citing the journal title: full title or abbreviated one, italics or not, ...etc.

→ In the references section, abbreviations and italics for journals have been revised and highlighted in red.

  1. also unify the way you write subtitles, first letter of every word either upper case or lower case in all subtitles.

→ The subtitles in the manuscript were reviewed, capitalized, and corrected. In addition, awkward expressions in the subtitles were corrected.

→ Other notes in the PDF file were also revised and highlighted in red. For example, UV detector, paragraph justified, etc.

→ [HPLC analysis was performed using an Agilent Infinity ll HPLC 1260 and UV detector.]

→ [In this study, we attempted to secure the analgesic effect and mechanism for chemotherapy-induced neuropathic pain based on the traditional efficacy of C. rotundus mentioned above. In addition, we tested the hypothesis that administration of C. rotundus and its metabolite α-cyperone would prevent and suppress paclitaxel-induced chemotherapy-induced peripheral neuropathic pain by modulating the norepinephrine analgesic system.]

Reviewer 2 Report (Previous Reviewer 1)

Comments and Suggestions for Authors

Dear authors,

thank you for taking into account the observations made and clarifying the indicated aspects.

My recommendation is at this time to accept in present form

Author Response

We thank you for accepting our paper and hope that our authors will also contribute to metabolites.

Thanky you very much.

This manuscript is a resubmission of an earlier submission. The following is a list of the peer review reports and author responses from that submission.

Round 1

Reviewer 1 Report

Comments and Suggestions for Authors

Dear authors,

The research undertaken is of major interest for the control of adverse reactions from antitumor chemotherapy.

Next, I will formulate the observations that justify the final recommendation.

The article requires a careful check of the English language used in the writing. Only few examples: C. Rodundus (line 200), ("and is excellent in hypotensive [23]")-line 56, etc.

Regarding the complex experiment carried out by the authors, I would like some clarifications:

- what are the other components of the vegetable product; is alpha-cyperone the most quantitatively expressed product from C. rotundus?

- are there no other compounds that could determine the supposed effect?

- how did the authors get the alpha-cyperone administered? How did they quantify the most effective dose?

- did the authors determine a toxic dose for the plant product and for alpha-cyperone?

- the authors postulate an interference of alpha-cyperone on beta tubulin. Did the authors also test a potential direct antitumor effect of this substance?

- where are the results of the docking study presented?

Under these conditions, my recommendation is to publish with major revisions after fixing these observations, which I would like to appear in the article as clarifications.

Comments on the Quality of English Language

The article requires a careful check of the English language used in the writing. Only few examples: C. Rodundus (line 200), ("and is excellent in hypotensive [23]")-line 56, etc.

Reviewer 2 Report

Comments and Suggestions for Authors

The work described in this manuscript is well prepared and described. Few mistakes, mainly linguistic, need to be corrected. Authors should:

- add a section for the abbreviations used in this manuscript.

- should provide the concentration of α-cyperone in the tested extract, better to expressed as mg/kg plant material of mg/g dry extract.

- it would be better to indicate the presence of any other constituents in the tested plant material.

authors should carefully revise the list of references and correct all indicated mistakes.

Comments on the Quality of English Language

some few typos and grammatical mistakes were noticed and are all indicated in the attached files